# Structure of the transcription activator target Tra1 within the chromatin modifying complex SAGA

Grigory Sharov [1,2,3,4], Karine Voltz[1,2,3,4], Alexandre Durand[5], Olga Kolesnikova[1,2,3,4], Gabor Papai[1,2,3,4], Alexander G. Myasnikov[6], Annick Dejaegere[1,2,3,4], Adam Ben Shem[1,2,3,4] & Patrick Schultz [1,2,3,4]

The transcription co-activator complex SAGA is recruited to gene promoters by sequence-specific transcriptional activators and by chromatin modifications to promote pre-initiation complex formation. The yeast Tra1 subunit is the major target of acidic activators such as Gal4, VP16, or Gcn4 but little is known about its structural organization. The 430 kDa Tra1 subunit and its human homolog the transformation/transcription domain-associated protein TRRAP are members of the phosphatidyl 3-kinase-related kinase (PIKK) family. Here, we present the cryo-EM structure of the entire SAGA complex where the major target of activator binding, the 430 kDa Tra1 protein, is resolved with an average resolution of 5.7 Å. The high content of alpha-helices in Tra1 enabled tracing of the majority of its main chain. Our results highlight the integration of Tra1 within the major epigenetic regulator SAGA.

[1] Department of Integrated Structural Biology, Institut de Génétique et de Biologie Moléculaire et Cellulaire, 67404 Illkirch, France. [2] Centre National de la Recherche Scientifique, UMR7104, 67404 Illkirch, France. [3] Institut National de la Santé et de la Recherche Médicale, U964, 67404 Illkirch, France. [4] Université de Strasbourg, 67404 Illkirch, France. [5] Research Group 'Chromosome Organization and Dynamics', Max Planck Institute of Biochemistry, Am Klopferspitz 18, 82152 Martinsried, Germany. [6] Department of Biochemistry and Biophysics, University of California San Francisco, San Francisco, CA 94158-2517, USA. Correspondence and requests for materials should be addressed to A.B.S. (email: adam@igbmc.fr) or to P.S. (email: patrick.schultz@igbmc.fr)

Transcription of protein coding genes by RNA polymerase II requires specific assembly of a pre-initiation complex (PIC) at gene promoters. Immediately before PIC assembly trans-activator proteins bind to specific DNA sequences in response to cellular cues and recruit co-activator complexes to alter chromatin structure and to coordinate transcription with epigenetic modifications[1,2]. The 19-subunit SAGA co-activator complex binds to activators in vivo[3,4] and harbors two enzymatic activities to acetylate or deubiquitinate nucleosomal histones[5,6]. The molecular basis for SAGA recruitment by activators and for relaying this event to its enzymatic centers is poorly understood. Currently available structural information on full SAGA is derived from low-resolution electron microscopy studies[7–9], crosslinked mass spectrometry,[10] and combinatorial depletion analysis[11]. A higher resolution structure is required to investigate its assembly, interactions between the different functional centers, and role in transcription regulation.

Here we present the cryo-EM structure of the entire SAGA complex where the major target of activator binding, the 430 kDa Tra1 protein is resolved with an average resolution of 5.7 Å. SAGA is organized into two lobes, one of which is fully occupied by Tra1 while the second contains enzymatic and chromatin recognition functions. The interaction between Tra1 and the second lobe relies on a single and narrow interface forming a hinge region. Tra1 is a member of the PIKK (PI3K-like kinases) family that contains three major domains: HEAT, FAT, and kinase[12,13]. We find a sequence motif that seems essential for the intricate HEAT domain architecture of Tra1, we identify a structural core shared by other PIKKs including TOR, and we illuminate a resemblance between Tra1 and DNA-PK, a PIKK involved in DNA repair[14]. In addition we show that the catalytically inactive kinase of Tra1 presents a much more accessible central cleft than true kinases. Our work uncovers design principles of large HEAT repeat domains in addition to providing a framework for studying the pivotal event of SAGA recruitment by activators.

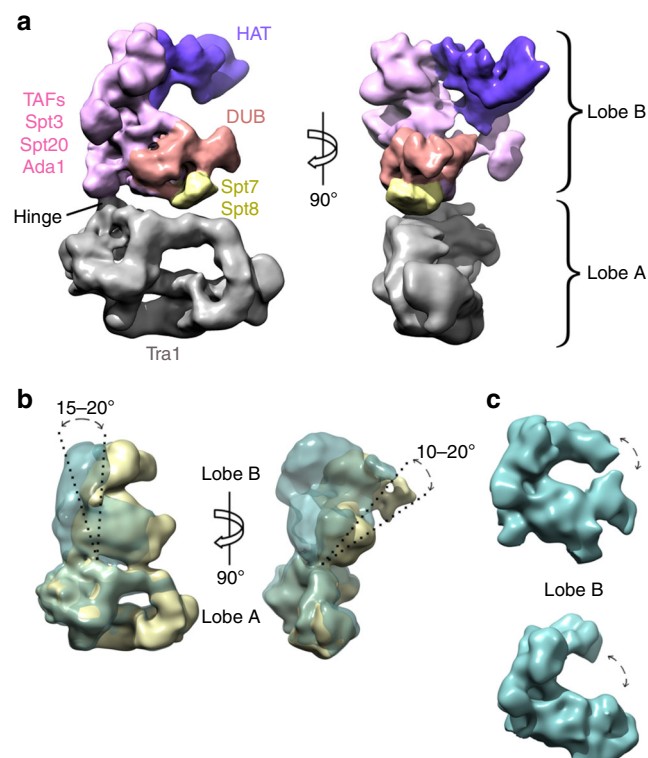

**Fig. 1** Cryo-EM structure of the *P. pastoris* SAGA complex. **a** Cryo-EM structure at 11.7 Å resolution of the full SAGA complex formed by lobe A (gray) and lobe B (colored). The approximate positions of key SAGA subunits, the histone acetyltransferase (HAT) and deubiquitination (DUB) modules are depicted according to literature[7–9]. **b** Analysis of SAGA complex flexibility by 3D classification showing a swirling of lobe B with respect to lobe A and **c** changes in the conformation of the HAT and DUB modules within lobe B. Scale bar represents 5.1 nm in **a** and 9 nm in **b** and **c**

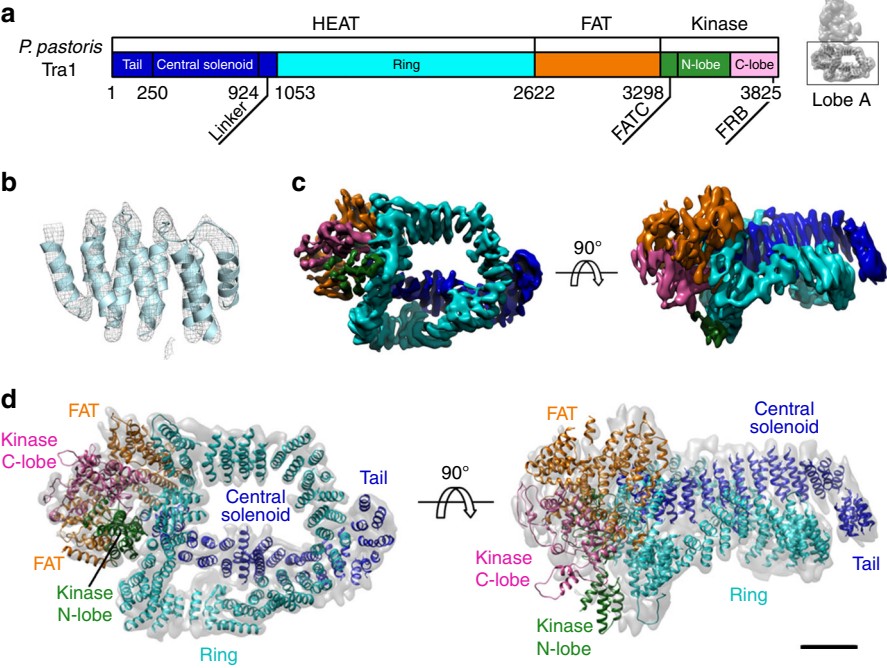

**Fig. 2** Architecture of the Tra1 subunit. **a** Structural units of the *P. pastoris* Tra1 subunit comprised of HEAT, FAT, and kinase domains. **b** Enlarged view of a segment of the HEAT domain showing the quality of the density map that reveals the α-solenoid path. **c** Cryo-EM map of SAGA's lobe A containing exclusively the Tra1 subunit, color-coded by structural domains as in **a**. **d** Main chain model of the Tra1 subunit

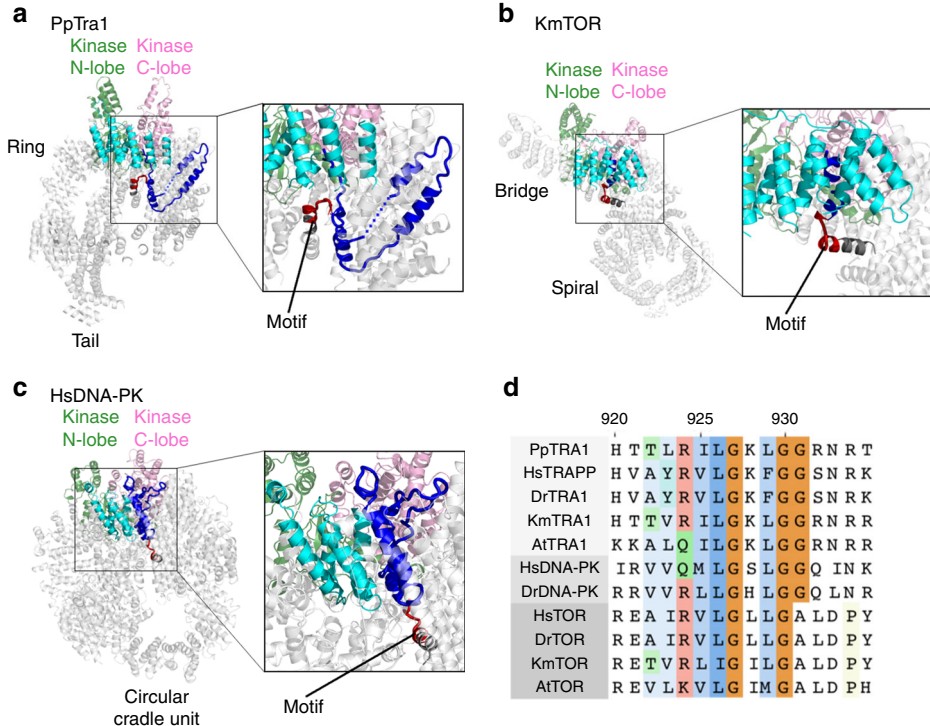

**Fig. 3** Common structural feature within the PIKK superfamily. Comparison of Pp Tra1 (**a**), Km TOR (**b**), and Hs DNA-PK (**c**) structures revealing a common structural motif (red), which ends the central solenoid in *P. pastoris* Tra1 (residues 924–933), the spiral domain in Km TOR (residues 850–860), and the N-terminal unit in Hs DNA-PK (residues 857–866). The linkers between the central solenoid and the ring in Tra1, between the spiral and bridge in TOR or between the circular cradle and the N-terminal unit are colored blue, core helices adjacent to the kinase domain appear in cyan, and the kinase domain is represented as in Fig. 2a. **d** Sequence alignment of the conserved structural motif [RKQC]-Ψ-Ψ-G-X-Φ-G (where Ψ represents amino acids with large aliphatic side chains, Φ any hydrophobic residue, and X any residue) placed at the end of the central solenoid. The represented species are *Homo sapiens* (Hs), *Pichia pastoris* (Pp), *Dario rerio* (Dr), *Kluyveromyces marxianus* (Km), and *Arabidopsis thaliana* (At). The sequence numbering refers to the sequence of PpTra1

## Results

**Structure determination of the SAGA complex.** There are only a few hundred copies of SAGA in a cell. To obtain sufficient amounts of homogenous sample, we developed a purification procedure for endogenous nuclear complexes that exploits the ability of the yeast *Pichia pastoris* to reach high cell densities while maintaining exponential growth rate (Supplementary Fig. 1). Single-particle analysis of mildly cross linked, frozen hydrated SAGA molecules resulted in a 3D reconstruction with an overall resolution of 11.7 Å (Supplementary Fig. 2; Supplementary Table 1). SAGA has an elongated shape of 28 × 20 nm in size and is formed by two asymmetric lobes connected by a single interaction site (Fig. 1a). Three-dimensional classification revealed a continuous twisting movement between the two lobes around this hinge region (Fig. 1b). Furthermore, an extended protein domain in the largest lobe B can swing by 15° from an open to a closed state (Fig. 1c). This flexibility is likely an important feature of SAGA to facilitate the search for a substrate nucleosome upon activator recruitment.

To overcome flexibility-induced blurring, we analyzed each lobe separately by a localized reconstruction approach[15]. While lobe B improved only modestly due to uncorrelated domain movements, the resolution in lobe A reached 5.7 Å, thus showing clear density for helices and several connecting loops (Fig. 2). Tra1 was previously localized in this lobe[7,9]. We now find that Tra1 occupies the full volume of lobe A, implying that SAGA activator recruitment and enzymatic functions are spatially separated.

We could trace most of Tra1 main chain and model the structured domains throughout the protein with the exception of some loops. The fitted crystal structures of the mTOR FAT and kinase domains[16] served as a starting point for constructing a model of the homologous domains in Tra1, where significant differences were observed (Supplementary Figs. 3, 4). In addition we built the ~300 kDa HEAT domain including all ~103 predicted alpha helices (Supplementary Figs. 4, 5). The HEAT repeats form a central solenoid overlaid by a ring structure and adopt a cradle shape when viewed from the side (Fig. 2c, d). In agreement with early antibody labeling experiments[7], the N-terminus of Tra1 is located in a short protruding tail containing 10 α-helices corresponding to residues 1–250. Extending the tail, the central solenoid comprises 11 regular HEAT repeats up to residue 928. These are followed by a linker domain containing three helices before assuming a regular ring-shaped, α-solenoid path, of 31 HEAT repeats (Fig. 3a). This ring (Fig. 2) crosses the central solenoid (at repeat 19) before merging into the FAT domain in a region close to lobe B. An intricate network of interactions is detected between the central and ring solenoids in the region where they cross each other. This might contribute to establishing or stabilizing the described architecture. The topology of the HEAT domain was validated by the good correlation between the predicted and the measured helix dimensions and by comparison with intra-subunit cross-linking/mass spectrometry experiments[10] (Supplementary Fig. 6).

**Homology with PIKK family members.** The large variability in their overall length and their low sequence homology suggest that PIKK HEAT domains may adopt diverse quaternary structures[12]. Sequence and secondary structure analyses predicted the occurrence of shared elements but these were never structurally identified[17]. The structural comparison of the HEAT domains of

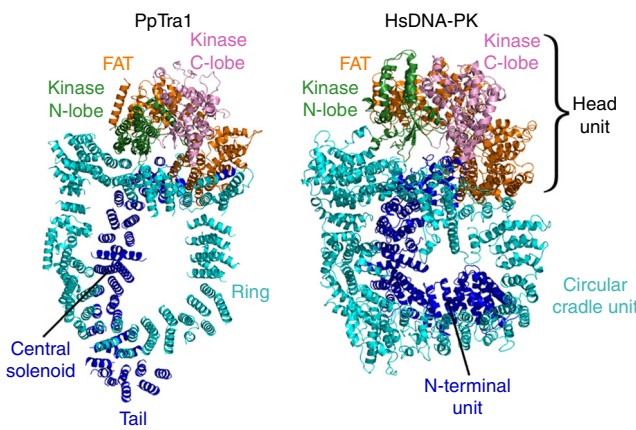

**Fig. 4** Structural homology between Tra1 and DNA-PK. Comparison of Tra1 and DNA-PK structures showing their similar architecture and domain organization

TOR, Tra1, and DNA-PK unveil common architectural principles, as well as shared structural elements. When the FAT-kinase domains are superposed, we find that the C-termini of the Tra1 central solenoid, the TOR spiral solenoid[18,19] and the mTOR N-terminal unit[20] occupy equivalent spatial positions, and share a common structural element corresponding to a helix-sharp turn-short helix structure (red motif in Fig. 3). This structural element breaks the regular HEAT repeat solenoids and marks the beginning of a linker domain (blue in Fig. 3) that reaches another stretch of HEAT repeats, respectively, the ring domain for PpTra1 (Fig. 3a), the bridge domain for KmTOR (Fig. 3b), and the circular cradle for HsDNA-PK (Fig. 3c). These solenoids are at very similar positions with respect to the kinase in all three proteins. Interestingly, this element also corresponds to the only sequence motif identified as strongly conserved between the HEAT domains of TOR, Tra1, and DNA-PK (Fig. 3d). Furthermore, immediately after this linker associated to the conserved motif, the eight helices closest to the kinase in the TOR HEAT domain have clear counterparts in Tra1 and in DNA-PK, and form a similar curvature in all proteins[18,19]. An even more comprehensive similarity is observed between Tra1 and DNA-PK (Fig. 4). Although the available model of DNA-PK does not contain the full set of HEAT repeats[20,21], the comparison with our Tra1 model shows that the overall architecture, shape, and size of the HEAT domain, as well as the positions of the FAT and kinase domains are almost identical. It is noteworthy that in Tra1 and DNA-PK, the HEAT repeats wrap around a much larger part of the kinase as compared to the shorter TOR and thus additional interactions are formed with the kinase. The striking similarity in overall architecture between DNA-PK and Tra1 suggests that the architecture of even very large HEAT domains can be dictated by a small number of crucially placed elements.

Contrary to TOR and DNA-PK, Tra1 is a pseudo-kinase[22] that lacks catalytic activity[23]. The kinase domain nevertheless maintains the archetypal two-lobe fold of PI3K and the loops that carry the ATP binding residues or the catalytic residues in TOR are located in almost identical positions in Tra1 (Fig. 5a). In contrast, regulatory elements that restrict substrate access to the catalytic cleft in TOR show strong differences in structure or orientation with respect to their Tra1 analogs. The four-helix bundle FRB occupies a completely different position and packs with the ring HEAT repeats in Tra1, the Tra1 activation loop has more than twice the length of its TOR counterpart and extends away from the central cleft and the LBE element also points away from the cleft in Tra1 (Fig. 5a). Strikingly, TOR's helix Kα9b and the negative regulatory loop sticks into the catalytic cleft and

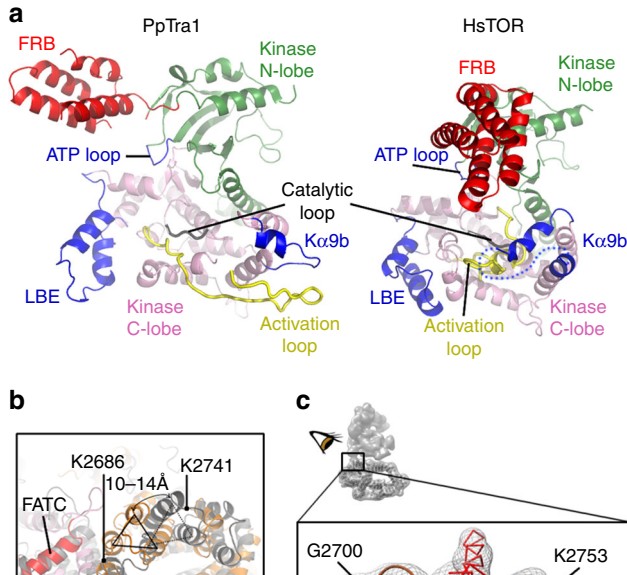

**Fig. 5** Unique features of the Tra1 FAT and Kinase domains. **a** Open conformation of Tra1 pseudokinase. Comparison of the kinase domain in *P. pastoris* Tra1 and mTOR (PDB 4jsnB). **b** Helices (indicated by a triangle, residues K2686–K2741) in the TRD2 domain of Tra1 FAT (orange) shift by up to 14 Å toward the kinase domain (pink) in comparison to mTOR (gray). The first helix of the FATC part of the kinase in Tra1 is in red. **c** Hinge interface between the two lobes. Backbone of a putative helix from lobe B in red. Helices from the FAT domain (residues G2700–K2753) forming the bearing-like surface in brown. The position of the hinge within the full SAGA complex is highlighted at the top left

plugs one end of it, while in Tra1 helix Kα9b proceeds in the opposite direction to interact with the FAT domain and there is virtually no analog for the negative regulatory loop. As a result the central cleft of the Tra1 kinase domain is much more accessible than in TOR.

Likewise, the FAT repeats surrounding the kinase domain display a similar overall organization with respect to other PIKKs, yet significant local differences are apparent. The FAT repeats are divided into three TRD (tetratricopeptide repeat-containing domain) domains and one HRD (HEAT repeat domain) domain[16]. The TRD2 domain in FAT is shifted much closer to the kinase forming new interactions notably with the kinase FATC domain known to be important for Tra1 assembly (Fig. 5b; Supplementary Fig. 4). Within the shifted TRD2 domain, three consecutive helices that are highly conserved among Tra1 proteins (residues 2698–2753) form a bearing-like module that accepts a rod-shaped density, probably an α-helix, originating from lobe B (Fig. 5c). This is the major, perhaps single, interface between the two lobes, though its surface area represents only a minute fraction of that presented by the two lobes (Figs. 5c, 1a). This interaction forms the hinge region that orchestrates the movement of one lobe with respect to the other (Fig. 1a). Cross-linking data suggest Taf12 and Spt20 as the most likely partners of Tra1 in the hinge region[10].

## Discussion

The present structural model of SAGA shows that Tra1 occupies entirely lobe A. This finding contradicts a previous model of NuA4 whose shape and size was highly similar to SAGA's lobe

A[24]. Since Tra1 is shared by SAGA and NuA4, this observation suggested that lobe A contains not only Tra1 but also the 9 NuA4-specific subunits. A high flexibility of the subunits associated with Tra1 in NuA4 or a partial dissociation may explain that only the Tra1 module was reconstructed in this study. We were not able to fit unambiguously any additional atomic structure in the enzymatic lobe. We tried to fit the previously determined 5TAF structure containing two copies of Taf5, 6, 9, 10, and 12 since this structure is partially shared between SAGA and TFIID[25]. We could not find such a highly symmetric substructure in the enzymatic lobe. This could be related to differences in subunit stoichiometry in SAGA vs. TFIID, which would indicate an alternative assembly mechanism. Indeed, while it is clearly demonstrated that a core of subunits is present in two copies in TFIID[26], such information is not available for SAGA.

Tra1 mutations that diminish SAGA's histone acetylation activity without affecting its assembly or its recruitment to chromatin, established the presence of a crosstalk between the enzymatic subunits and the activator binding sites, which are located in different lobes[20,27]. Intriguingly, Tra1 is recruited to promoters by numerous activators with distinct binding sites along the HEAT domain[9,27], but employs a single narrow interface for communicating this information to the enzymatic lobe (Figs. 5c, 1a; Supplementary Fig. 7).

The current view of PIKK regulation posits that various inputs, such as activating proteins binding to the HEAT domains or post-translational modifications, are integrated by PIKK to modify its kinase accessibility[16,28]. Tra1, however, is the only PIKK with a catalytically inactive kinase and its pseudo-kinase center is widely open. It is tempting to suggest that in Tra1, unlike all other PIKKs, the combined effect of various stimuli is integrated, not by the kinase, but into a conformation change within the FAT hinge module that might be further propagated to the enzymatic sites in lobe B. Tra1 is also part of the essential multi-protein co-activator NuA4[29] and as such mediates recruitment to the chromatin of additional distinct enzymatic activities as well as histone exchange machineries[30]. The structure presented here will facilitate uncovering the unique Tra1 features that made evolution select the same protein for the recruitment of different transcription regulatory activities. Understanding the molecular crosstalk between SAGA's biological functions will require additional structures with bound activators and nucleosomes as well as a better definition of the enzymatic lobe.

## Methods

**Preparative scale production of SAGA.** The SAGA complex was purified from nuclear extracts of a budding yeast *P. pastoris* (*Komagataella pastoris*) strain using a streptavidin-binding peptide (SBP) affinity tag placed at the C-terminus of the Sgf73 subunit (Supplementary Fig. 1). About 2 l of yeast cells were grown at 24 °C with glycerol as carbon source and harvested when OD$_{600\,nm}$ reached 12–15. Cells were washed in water and then treated with 10 mM DTT. The cell wall was digested by addition of lyticase and spheroplasts were pelleted at 5500×g for 20 min. All further steps were performed at 0–4 °C. Protease inhibitors were added to all buffers. Spheroplasts were disrupted by suspension in a hypotonic buffer (15–18% Ficoll 400, 0.6 mM MgCl$_2$, 20 mM K-phosphate buffer pH 6.6) using a dounce homogenizer. Sucrose (0.1 M) and MgCl$_2$ (5 mM) were then added. Nuclei (and some debris) were pelleted at 33,000×g for 33 min, resuspended in a wash buffer (0.6 M sucrose, 8% PVP, 1 mM MgCl$_2$, 20 mM phosphate buffer pH 6.6) and pelleted again at 34,000×g for 50 min. Nuclei were resuspended in extraction buffer (20 mM HEPES pH 8.0, 300 mM NaCl, 20% sucrose, 8 mM MgCl$_2$, 4 mM DTT) with 20 strokes using a tight pestle in a dounce homogenizer. Following 30 min of incubation, debris were precipitated at 35,000×g for 38 min. The supernatant was collected and 1–2% PEG 20,000 added in order to precipitate some remaining organelles and membrane parts by a short centrifugation step at 33,000×g for 10 min. The PEG 20,000 concentration was then increased to 5.8% and SAGA precipitated in a second short centrifugation step. The pellet was solubilized in a minimal volume and avidin was added to block endogenously biotinylated proteins. The suspension was incubated with streptavidin beads for 4 h in buffer A (20 mM HEPES pH 8.0, 250 mM NaCl, 10% sucrose, 2 mM MgCl$_2$, 2 mM Tris (2-carboxyethyl)phosphine) (TCEP) washed five times and eluted with buffer A

containing 10 mM biotin. The eluate was concentrated with Millipore Amicon-Ultra (50 kDa cutoff) and spun in a 10–30% sucrose gradient with buffer B (20 mM HEPES pH 8.0, 150 mM potassium acetate, 2 mM TCEP, 3 mM MgCl$_2$) in rotor SW60 (38,000 rpm for 13.5 h). SAGA was fractionated at ~25% sucrose and concentrated with Amicon-Ultra to ~1 mg per ml.

**Cryo-EM sample preparation and data acquisition.** Preliminary analysis of purified complex by negative staining and cryo-EM indicated that SAGA is very fragile and prone to aggregation and dissociation. By systematic screening of several detergents and buffer components, together with mild cross-linking of concentrated SAGA samples, we were able to reduce aggregation and stabilize the complex. Furthermore, we noticed that SAGA adopts a preferential orientation on the supporting carbon film due to its elongated shape and a preferred binding interface. To overcome this problem we used holey carbon grids and succeeded in isolating a homogeneous SAGA population suitable for single-particle cryo-EM analysis (Supplementary Fig. 2).

Freshly purified SAGA complexes were precipitated with PEG 20,000 to remove sucrose and suspended at a concentration of 0.2 mg/ml before cross-linking with glutaraldehyde (final concentration 0.1%) for 30 min on ice. About 3 µl of sample was applied onto a holey carbon grid (Quantifoil R2/2 300 mesh, in-house prepared carbon) rendered hydrophilic by a 20 s glow-discharge in air (2 mA current at $1.8 \times 10^{-1}$ mbar). The grid was blotted for 1 s (blot force 5) and flash-frozen in liquid ethane using Vitrobot Mark IV (FEI) at 4 °C and 95% humidity.

Images were acquired on a C$_s$-corrected Titan Krios (FEI) microscope operating at 300 kV in nanoprobe mode using the EPU software for automated data collection. Movie frames were collected on a 4k×4k Falcon II direct electron detector at a nominal magnification of 59,000, which yielded a pixel size of 0.11 nm. Eight movie frames were recorded at a dose of 7 electrons per Å$^2$ per frame corresponding to a total dose of 60 electrons per Å$^2$, but only the 7 last frames were kept for further processing.

**Image processing.** Movie frames were aligned, dose-weighted, and averaged using Motioncor2[31] to correct for beam-induced specimen motion and to account for radiation damage by applying an exposure-dependent filter. Movies with large global frame shifts were excluded from further analysis. Unweighted movie sums were used for contrast transfer function (CTF) estimation with CTFFIND4[32] or Gctf program[33], while dose-weighted sums were used for all subsequent steps of image processing. After manual screening, images with poor CTF, particle aggregation, or ice contamination were discarded. About 6000 SAGA particles were picked manually using the e2boxer program of EMAN2[34] and subjected to 2D classification in Relion[35]. Representative class average images showing SAGA in different orientations were then used as references for auto-picking with Gautomatch (http://www.mrc-lmb.cam.ac.uk/kzhang/Gautomatch/). Several cycles of automatic picking followed by 2D classification were performed, yielding a data set of 264,901 particles. This data set was analyzed in Relion 1.4 according to standard protocols. Briefly, the images were subjected to 2D classification to eliminate remaining bad particles. The structure was refined using a low-pass filtered starting model obtained previously by random conical tilt method[7]. Global resolution estimates were determined using the FSC = 0.143 criterion after a gold-standard refinement. Local resolution was estimated with ResMap[36].

Three-dimensional classification of the entire data set could not clearly separate distinct conformations of SAGA complex. Therefore we carried out a focused refinement of lobe A using the masked lobe A as a reference followed by a 3D classification without alignment. Resulting classes revealed a continuous set of lobe B conformations indicating a twisting movement between the lobes. The same analysis was performed on lobe B, which uncovered additional conformational changes involving the histone acetyl transferase and the deubiquitination modules.

Localized reconstruction of separate SAGA lobes was performed using the Localrec scripts[15]. First, particle coordinates were recalculated by applying shifts from the latest 3D refinement. New origin and orientations for each lobe of the complex were calculated and used to extract subparticles from the full particle images. Additionally, the unwanted densities were subtracted from subparticle images using the relion_project command of Relion[37]. The produced subparticles were further treated as regular single particles. Three-dimensional reconstruction, structure refinement, 3D classification, and post-processing were carried out in Relion. Illustrations were prepared with GIMP, the Chimera visualization software[38], and the PyMOL Molecular Graphics System, Version 1.8 Schrödinger, LLC.

**Tra1 sequence analysis and secondary structure prediction.** The sequence analysis and secondary structure predictions were performed using the *P. pastoris* Tra1 (PpTra1) sequence Uniprot ID F2QQ15. A multiple sequence alignment (MSA) was realized to identify conserved regions within the Tra1 HEAT domain as well as for secondary structure prediction of the HEAT domain (see below). Homologous sequences of PpTra1 were retrieved from the RefSeq database[39] with the blastp tool[40] on the NCBI website. All homologous sequences with an identity above 25% were retained, and the MSA was performed using Cobalt[41]. The alignment was manually curated in Jalview[42], and contained 86 sequences (41 vertebrates, 7 insects, and 38 fungus) in its final form. The same protocol was used

to build MSA of TOR and DNA-PK, using the *Homo sapiens* sequences as input (Uniprot P42345 and P78527 for TOR and DNA-PK, respectively). The MSA of TOR includes the same species diversity as that of Tra1, while DNA-PK was present only in metazoans.

Conserved sequence motifs shared by the HEAT domains of the three PIKK were searched using the alignment conservation score[43]. Specifically, we retained motifs containing at least five consecutive residues with conservation scores between 11 (maximum value) and 9[42]. A consensus secondary structure prediction of the HEAT domain of PpTra1 was built using four individual predictions obtained using the programs JPred[44], SPIDER2[,45], and HHpred[46], using either the sequence of PpTra1 or the previously built MSA as input. The predicted lengths of helices were compared to the lengths of helices fitted in the density map. The coherence of the predicted loop length with the observed density was also examined.

**Tra1 model building.** Alpha helices were initially detected and constructed with phenix.find_helices_strands[47]. The extent of the HEAT domain became clearly defined in this manner. Building of additional HEAT domain helices as well as correction of helices length and orientations were performed in Coot[48]. The number and size of the helices are in very good agreement with the predicted values and in many cases the connecting loops between successive helices were distinguishable allowing sequence assignment of the HEAT domain helices. However, at the current resolution, no density for protein side chains is discerned, so this assignment may hold some inaccuracies.

Initial rigid body docking of the human mTOR FAT domain (PDB 4jsn[16]) into the cryo-EM map of SAGA's lobe A was performed using the colores tool of Situs software[49]. Beforehand, the cryo-EM map was low-pass filtered to 7.5 Å and the density attributed to the Tra1 HEAT domain was removed. Additionally, we removed all loops from this homologous FAT domain and replaced the sequence with polyalanine residues. Angular sampling of 10° and standard cross-correlation method was used for exhaustive 6D positional search. A top scoring solution was found to be in close agreement with previous manual docking. During the docking, the distinct curvature of the FAT domain structure indicated that the original 3D reconstruction had the wrong hand and at this point we have flipped the map. The model was manually corrected according to density taking into account the secondary structure prediction as obtained from HHPRED.

Approximate location of the kinase domain was deduced by superposing the FAT-kinase structure of human mTOR (PDB 4jsn) on Tra1 FAT domain. The starting kinase model was further fitted into EM density by fit in map tool of Chimera. The model was manually corrected according to density taking into account the secondary structure prediction. In the vast majority of cases, the secondary structure prediction (e.g., positions and lengths of helices) was in excellent fit and we believe that errors in residue register in the FAT and kinase domains are minimal.

Local rigid body and real-space refinement were performed in Coot to better place secondary structure elements into the map. Model geometry was then idealized using phenix.geometry_minimization with secondary structure restraints.

**Data availability.** The cryo-EM maps have been deposited in the 3D-EM database (EMBL-European Bioinformatics Institute, Cambridge, UK) with accession codes EMD-3790 and EMD-3804 for Tra1 and full SAGA, respectively. The model coordinates of Tra1 were deposited in the PDB database with accession code 5OEJ. All other data and request for materials are available from the corresponding authors.

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

## Acknowledgements

We thank L. Bianchetti and R. Stote for helpful discussions. We acknowledge support from the Institut National de la Santé et de la Recherche Médicale (INSERM), the Centre National pour la Recherche Scientifique (CNRS), the Association pour la Recherche sur le Cancer (ARC), and the Fondation pour la Recherche Médicale (FRM). This work was supported by the ANR SAGA2, the ANR-IAB-2011-BIP:BIP [ANR-10-BINF-0003]; the grant ANR-10-LABX-0030-INRT, a French State fund managed by the Agence Nationale de la Recherche under the frame program Investissements d'Avenir ANR-10-IDEX-0002-02, the French Infrastructure for Integrated Structural Biology (FRISBI) [ANR-10-INSB-05-01] and INSTRUCT as part of the European Strategy Forum on Research Infrastructures (ESFRI).

## Author contributions

P.S. and A.B.S. designed the study; A.B.S. originated the purification procedure; A.B.S. and O.K. produced, purified, and characterized the *P. pastoris* SAGA complex; A.B.S. treated purified SAGA before deposition on grids. G.S., G.P., and A.G.M. froze grids, collected and analyzed cryo-EM data, calculated and refined the EM density; K.V. and A.D. carried out secondary structure predictions and identified conserved motifs. P.S., K.V., G.S., and A.B.S. interpreted the structure by fitting crystal coordinates and model building. P.S. and A.B.S. supervised the work. G.S., P.S., A.B.S., K.V., and A.D. prepared figures and wrote the manuscript together.

## Additional information

**Competing interests:** The authors declare no competing financial interests.

