## [Peer Review File · Nature Communications]

Reviewers' comments:

Reviewer #1 (Remarks to the Author):

This manuscript describes the cryo EM structure of the yeast coactivator SAGA. This structure is a major advance in understanding the structure and function of SAGA, as it is of much higher resolution than earlier studies. In this work, the authors focus on SAGA lobe B, the region containing the Tra1 subunit. The resolution of the EM allows the authors to model the Tra1 polypeptide chain giving the overall architecture of this PIKK-like molecule and how it interacts with the rest of SAGA. Importantly, this also allows understanding how Tra1 relates to other PIKK family members. They find a striking similarity with the HEAT repeat regions of other factors as well as significant differences in the kinase-like domain that are unique to Tra1. Overall, this is a high impact paper. Suggestions for improvement are below.

1) Pg 4, paragraph starting with: The large variability in their overall.....

This paragraph and the related Fig 3 describe common architectural principles shared between HEAT domains of PIKK family members. The paragraph as written is unclear and the associated figure and legend are also unclear. Also, Fig 3 can be improved by larger and better labeled figures. For example, the key conserved region in Fig 3b is barely visible in Fig 3a and not labeled. Zooming in on the key regions discussed in Fig 3a and 3c are needed.

2) Pg 6, paragraph starting with: likewise, the Fat repeats...

What is the TRD2 domain being discussed? Some explanation is required. Also, TRD3 is not labeled well in Fig S3 – is it just one helix?

3) Fig S3: the resolution of this figure is not sufficient to read residue numbers or amino acid identity.

Reviewer #2 (Remarks to the Author):

The paper reports the cryo-EM structure of the yeast *Pichia pastoris* SAGA chromatin modifying complex. Due to flexibility inherent, the authors could only obtain good resolution out of a section of the complex. The main new result presented in the paper is therefore the structure of the large subunit of SAGA, Tra1, and its location within the complex, derived from a locally refined ~6 Å resolution cryo-EM map. The authors show that Tra1 closely resembles DNA-PK, which, like Tra1, belongs to the PIKK family.

On the technical side I am concerned with the lack of figures showing details about the modelling. The only place where the authors show their atomic model docked into their map is in Fig 2b, and they only show a few helices fit into a selected portion of the density. Since this model is the only result of the paper, it would have been nice to have an idea of how well the model fits the map throughout, and how close their built model is to their starting homology models (i.e. if large portions of the homologous xtal structures can be docked directly into the map, then one could have more confidence in those parts of the model). It would also be good to see the density map for the important regions that are highlighted in the text, e.g. in Fig 1a. It is sort of acceptable now to only show the atomic model for high-resolution EM structures, but for a map with a resolution of ~ 6Å interpretations made at the atomic/residue level should be clearly supported by the density. While it may be the case here, the present manuscript does not prove it.

The authors concentrate on Tra1 and ignore the rest of the structure. Since the SAGA complex should contain most of the subunits in the 5TAF/7TAF model that the authors presented previously for a subcomplex of TFIID (TAF5, 6, 9, 10 and 12, and subunits homologous to TAF4 and TAF8), I think they should minimally comment on whether those models or EM densities are consistent with

any of the density in the top lobe of their SAGA reconstruction.

Finally, my main issue with the paper is that there is limited new biological insight. An obvious thing this story is missing to have more impact is additional structure(s) with some bound substrate (e.g. nucleosome, activator, or DNA).

Reviewer #3 (Remarks to the Author):

This report builds on previous single-particle EM studies of the SAGA co-activator complex by Dr. Schultz and his colleagues. The overall resolution of the SAGA cryo-EM map is relatively low (~1.2 nm) by present day standards, due to a high degree of conformational variability. However, advances in EM image quality and image analysis algorithms allowed the authors of the study to determine a much more detailed (0.6 nm resolution) structure of the 430 kDa Tra1 PIKK-family protein, which forms the largest of the two lobes apparent in the overall SAGA structure.

A high proportion of alpha-helical elements in Tra1, and consideration of higher-resolution structures of homologous HEAT, FAT and kinase domains made possible tracing the entire Tra1 main chain in the corresponding cryo-EM map. The authors provide a detailed description of Tra1's intricate secondary structure organization, cross-validate their model by considering results from published cross-linking/MS analysis and discuss differences and similarities with other PIKK-family proteins, notably TOR and DNA-PK.

In contrast with the detailed description of the Tra1 lobe, this study provided little new insight about the organization of SAGA's catalytic lobe. This was due to limited resolution of that portion of the cryo-EM map. Nonetheless, the study identified the Tra1 domains directly involved in interaction with the catalytic lobe and suggested how the organization of the lobe interface might facilitate repositioning of the catalytic domain.

It is unfortunate that intermediate resolution information was limited to the Tra1 component and that the cryo-EM map could not provide more information about the organization of catalytic domains. Nonetheless, a detailed description of the Tra1 structure is an important step forward. A few points that perhaps could be better addressed in the manuscript:

- 1) The authors refer in various places to a "flow" of information from Tra1 to the catalytic lobe through the lobe interface. However, the authors do not explain how it has been established that such flow must exist. Can the possibility that interaction of activators with Tra1 simply recruits catalytic activities but does not modify or control them?
- 2) A published cryo-EM study of the NuA4 HAT complex from Chittuluru et al. suggested that Tra1 also forms an entire half of the NuA4 structure. Could the authors comment on the possible significance of this observation? Could Tra1 function not just as a target for activator binding, but perhaps making a contribution through its structure to facilitate interaction with a chromatin substrate?
- 3) The authors suggest that variability in the relative position of SAGA's Tra1 and catalytic lobes could facilitate interaction with a nucleosome substrate. Is there specific evidence to support this suggestion, or is the comment just a plausible way to rationalize the observed structural flexibility in the SAGA complex?
- 4) A sentence in the Abstract indicates that "The molecular basis for SAGA recruitment by activators and for relying this event to its enzymatic centres is poorly understood". It seems that neither of these points are really addressed by the results presented in this study.

Reviewer #1:

1) Pg 4, paragraph starting with: The large variability in their overall..... This paragraph and the related Fig 3 describe common architectural principles shared between HEAT domains of PIKK family members. The paragraph as written is unclear and the associated figure and legend are also unclear. Also, Fig 3 can be improved by larger and better labeled figures. For example, the key conserved region in Fig 3b is barely visible in Fig 3a and not labeled. Zooming in on the key regions discussed in Fig 3a and 3c are needed.

On behalf of the reviewer's comments we split figure 3 into two independent figures to better convey the information about the conserved motif (revised figure 3) and the homology with human DNA-PK (revised figure 4). We improve labelling and identified clearly the conserved motif in the figures. During the revision period an improved structure of DNA-PK was published by the Blundell team (Sibanda et al. Science. 2017 Feb 3;355(6324):520-524.) and the revised figures 3 and 4 now include this new structure. The paragraph on page 4 was also rewritten in a clearer way.

2) Pg 6, paragraph starting with: likewise, the Fat repeats...What is the TRD2 domain being discussed? Some explanation is required. Also, TRD3 is not labeled well in Fig S3 – is it just one helix? *The FAT repeats have been divided into 3 TRD (TPR tetratricopeptide repeat-containing domain) domains and one HRD (HEAT Repeat Domain) domain according to the nomenclature proposed by Yang et al., (Nature. 2013 May 9;497(7448):217-23). This abbreviations are now described in the text and the boundaries of these domains, shown in fig.S3, have been reinforced for sake of clarity.*

3) Fig S3: the resolution of this figure is not sufficient to read residue numbers or amino acid identity.

The resolution of this figure was improved in the revised version in order to allow clear visualization of the residue number and identity.

Reviewer #2:

I am concerned with the lack of figures showing details about the modelling. The only place where the authors show their atomic model docked into their map is in Fig 2b, and they only show a few helices fit into a selected portion of the density. Since this model is the only result of the paper, it would have been nice to have an idea of how well the model fits the map throughout, and how close their built model is to their starting homology models (i.e. if large portions of the homologous xtal structures can be docked directly into the map, then one could have more confidence in those parts of the model). It would also be good to see the density map for the important regions that are highlighted in the text, e.g. in Fig 1a. It is sort of acceptable now to only show the atomic model for high-resolution EM structures, but for a map with a resolution of ~ 6Å interpretations made at the atomic/residue level should be clearly supported by the density. While it may be the case here, the present manuscript does not prove it. *To show the good fit of our model with the cryo-EM map, we now provide a revised figure 2d in*

which the contoured density map is superimposed to the model. Furthermore we provide two new supplemental figures (figure S3 and S4) showing the original kinase (mTOR) and FAT (mTOR) before and after adapting the atomic model to the EM density as well as the modelling of the HEAT repeats. These new figures show better the different steps in the modelling and the good fit between our model and the density map.

The authors concentrate on Tra1 and ignore the rest of the structure. Since the SAGA complex should contain most of the subunits in the 5TAF/7TAF model that the authors presented previously for a subcomplex of TFIID (TAF5, 6, 9, 10 and 12, and subunits homologous to TAF4 and TAF8), I think they should minimally comment on whether those models or EM densities are consistent with any of the density in the top lobe of their SAGA reconstruction. *We were not able to dock unambiguously any structure in the enzymatic lobe. We obviously tried to fit the previously determined 5TAF structure containing two copies of TAF5, 6, 9, 10 and 12. At this stage we could not find such a highly symmetric sub-structure in the enzymatic lobe. This could be related to differences in subunit stoichiometry in SAGA vs TFIID which would indicate an alternative assembly mechanism. Indeed, while it is clearly demonstrated that a core of subunits is present in two copies in TFIID, such information is not available for SAGA. Our semi-quantitative mass spectrometry data do not show any evidence for an increased copy number of any subunit. This point is now included in the discussion on page 6*

Finally, my main issue with the paper is that there is limited new biological insight. An obvious thing this story is missing to have more impact is additional structure(s) with some bound substrate (e.g. nucleosome, activator, or DNA).

Our preliminary structural experiments aiming at identifying the binding site of activators within the structure of Tra1 were unsuccessful, although biochemical control experiments proved activator binding. The difficulty to reveal the bound activators might be due to the small size and flexible nature of the acidic activating domains or to possible multiple binding sites. Additional work is needed to sort out this essential question.

Reviewer #3:

1) The authors refer in various places to a “flow” of information from Tra1 to the catalytic lobe through the lobe interface. However, the authors do not explain how it has been established that such flow must exist. Can the possibility that interaction of activators with Tra1 simply recruits catalytic activities but does not modify or control them?

The most striking clue for an allosteric mechanism linking SAGA recruitment and the catalytic activities of SAGA comes from genetic experiments showing that point mutations or deletion mutants in Tra1, especially in the pseudo-kinase region, affect the Histone acetyl transferase activity without altering promoter recruitment of SAGA. This point is discussed on page 7. In agreement with the

referee we removed this concept from the introduction since it was not clearly explained at that stage.

2) A published cryo-EM study of the NuA4 HAT complex from Chittuluru et al. suggested that Tra1 also forms an entire half of the NuA4 structure. Could the authors comment on the possible significance of this observation? Could Tra1 function not just as a target for activator binding, but perhaps making a contribution through its structure to facilitate interaction with a chromatin substrate?

The structure of Tra1 within SAGA corresponds to the cryo-EM structure of NuA4 that was reported previously. We are currently investigating the structure of NuA4 and the most likely explanation is that NuA4 was dissociated in the experiments reported by Chittuluru et al. SAGA is also extremely fragile and especially this interface between the two lobes appears labile. We had to cross-link the SAGA sample to preserve this interaction in most of the molecules and we suspect that a similar dissociation occurs in NuA4. We discuss this point respectfully on page 6 suggesting that in that early study the NuA4-specific subunits may be highly flexible or sub-stoichiometric.

We currently have little information about any contribution of Tra1 in chromatin recognition. All known chromatin interaction domains or epigenetic readers are placed in the upper lobe. Our low resolution investigation of the SAGA-nucleosome interaction did not reveal any significant binding of nucleosomes to the Tra1 lobe.

3) The authors suggest that variability in the relative position of SAGA's Tra1 and catalytic lobes could facilitate interaction with a nucleosome substrate. Is there specific evidence to support this suggestion, or is the comment just a plausible way to rationalize the observed structural flexibility in the SAGA complex?

The catalytic domains, and in particular the HAT module, are highly flexible. No specific evidence is available to demonstrate that this flexibility has any biological role and we express a plausible hypothesis suggesting that these modules act as mobile reading heads to find their substrate. Further experiments with bound nucleosomes will provide more accurate information.

4) A sentence in the Abstract indicates that "The molecular basis for SAGA recruitment by activators and for relying this event to its enzymatic centres is poorly understood". It seems that neither of these points are really addressed by the results presented in this study

This sentence was removed from the abstract and is now in the introduction. Additional work is needed to sort out these essential questions. We however feel that the sentence is still valid and that the detailed mechanism remains to be solved.

REVIEWERS' COMMENTS:

Reviewer #1 (Remarks to the Author):

The manuscript has been revised to address my comments and I recommend publication. It is an important advance in understanding the function of the SAGA complex and the structure and role of the conserved PIKK-like subunit Tra1.

Reviewer #2 (Remarks to the Author):

The authors have done the best they presently can to address my comments. It is obviously disappointing that no new information has been obtained for Lobe B, in spite of the author's previous work concerning some of the involved TAFs (even if assuming a single copy for each).

The docking of the crystals structures is convincing and better demonstrated in the new version.

One last comment, but it is what it is, is that the overall resolution for Lobe A as determined by the FSC curve seems an overestimation, specially in the context of the local resolution map, which is mostly green and cyan, indicating something closer to 7.5 than 5.7 Å. This could be due to the effect of the mask. This reviewer is not particularly concerned about this, except that it appears in the abstract as a matter of fact. The authors may decide what to do about it. Some in the cryo-EM community would have more of an issue than I do.

I believe that the paper is of value and is ready for publication.

Reviewer #3 (Remarks to the Author):

The authors made an honest effort to address concerns expressed by the reviewers. Some of their answers mostly acknowledge that the results presented in the manuscript are not sufficient to resolve the issues that were raised. There are several essential questions about SAGA structure and mechanism that are not answered by the results described in the manuscript, but the results at hand are described appropriately

Reviewer #2 (Remarks to the Author):

The authors have done the best they presently can to address my comments. It is obviously disappointing that no new information has been obtained for Lobe B, in spite of the author's previous work concerning some of the involved TAFs (even if assuming a single copy for each).

The docking of the crystals structures is convincing and better demonstrated in the new version.

One last comment, but it is what it is, is that the overall resolution for Lobe A as determined by the FSC curve seems an overestimation, specially in the context of the local resolution map, which is mostly green and cyan, indicating something closer to 7.5 than 5.7 Å. This could be due to the effect of the mask. This reviewer is not particularly concerned about this, except that it appears in the abstract as a matter of fact. The authors may decide what to do about it. Some in the cryo-EM community would have more of an issue than I do.

We do recognize that the local resolution figure (supplemental Figure 2e) does not perfectly reflect the values obtained by the Fourier Shell Correlation plot. We experienced several times that the resolution values provided by the local resolution map are slightly worse than the one calculated from the FSC curve. One possible reason is that we represent the surface of the structure which is slightly more mobile than its core. The local resolution map is generally used to detect mobile parts of the structure and is not used as a resolution criteria by the cryo-EM community. The most widely accepted resolution value is the one provided by the FSC curve and when the structures of the two half datasets were determined independently. We therefore suggest to keep this value in the text as it is an internationally recognized criteria in the field. The question raised by the referee about the masking of the structure for resolution determination is of importance since a too tight mask may artificially improve the resolution value. We have used a wide mask with a 6 pixel extension from the density threshold and a 6 pixel Gaussian fall-off. This total of 12 pixels extension is very generous and meets the expectations used in the field.